# Site Matters: Differences in Gene Expression Profiles Along the Bovine Rumen Papilla During Subacute Rumen Acidosis

**DOI:** 10.3390/ijms252212303

**Published:** 2024-11-16

**Authors:** Arife Sener-Aydemir, Franziska Dengler, Filip Larsberg, Raul Rivera-Chacon, Ezequias Castillo-Lopez, Qendrim Zebeli, Susanne Kreuzer-Redmer

**Affiliations:** 1Centre for Animal Nutrition and Animal Welfare Sciences, University of Veterinary Medicine Vienna, Veterinärplatz 1, 1210 Vienna, Austria; arife.seneraydemir@gmail.com (A.S.-A.); filip.larsberg.1@hu-berlin.de (F.L.); raul.rivera-chacon@vetmeduni.ac.at (R.R.-C.); ezequias.castillo-lopez@vetmeduni.ac.at (E.C.-L.); qendrim.zebeli@vetmeduni.ac.at (Q.Z.); 2Institute of Physiology, Pathophysiology & Biophysics, University of Veterinary Medicine Vienna, Veterinärplatz 1, 1210 Vienna, Austria; franziska.dengler@uni-hohenheim.de; 3Institute of Animal Science, University of Hohenheim, 70599 Stuttgart, Germany

**Keywords:** rumen epithelium, gene expression, mucosal barrier, inflammation, nutrient transport

## Abstract

Subacute rumen acidosis (SARA) is a significant concern in dairy cattle fed grain-rich diets. To elucidate the underlying pathophysiological mechanisms, ruminal papilla biopsies are often used. This study aimed to assess how the sampling site along the ruminal papilla influences gene expression profiles in rumen epithelium during SARA. Rumen biopsies from five ruminal-cannulated non-lactating Holstein cows were collected during feeding of a forage diet (FD) and seven (wk1) and 21 days (wk3) after transition to high-grain (HG) feeding. Gene expression in apical (AP), basal (BP), and total length (TP) papillae were compared using RT-qPCR. Significant diet-induced effects were observed in AP for *DSG1* (wk3, *p* = 0.0317), *ZO1* (wk1 and wk3, *p* = 0.0159), *GLUT3* (wk3, *p* = 0.0159), *TLR4* (wk3, *p* = 0.0635), and *NFKB* (wk1, *p* = 0.0159), but hardly in BP or TP. Within wk1, TP showed higher transcript levels of *ZO1* and *TLR4* (*p* = 0.0079) and *SGLT1* (*p* = 0.0317) compared to AP and BP independently from diet effects. These findings suggest that the apical parts of rumen papillae biopsies are most suitable for gene expression analyses to investigate diet-induced effects on rumen physiology and underscore the importance of considering the sampling site for accurate gene expression studies in rumen epithelium during SARA, providing valuable insights for future research and diagnostic approaches in managing rumen health in dairy cattle.

## 1. Introduction

The rumen is a complex and dynamic microbial ecosystem domiciling bacteria, protozoa, fungi, archaea, and viruses. Such microbial consortia promote the efficient utilization of feeds rich in structural carbohydrates by fermentation into volatile fatty acids (VFA), the major source of energy for ruminants, which are absorbed by the stratified squamous rumen epithelium [1,2]. Thus, optimal rumen functionality is crucial for the efficient energy supply, epithelial integrity, and overall health of the animal. This includes the maintenance of ruminal pH [3], which is subject to diet composition and resulting fermentation processes.

While grain-rich diets in dairy cattle are high-energy sources, which support and foster high milk production [4,5,6], increased consumption of highly fermentable carbohydrates may lead to the accumulation of ruminal VFA and reduction in ruminal buffering capacity [4,7,8,9]. The resulting drop in pH can result in subacute ruminal acidosis (SARA), inducing dysbiosis and disrupting epithelial key functions, i.e., (1) barrier formation, (2) nutrient uptake, and (3) pathogen exclusion and immune response [10,11,12,13,14,15]. It is widely accepted that alterations in papillae proliferation, density, or size correspond to the diet, especially to the intake of proteins and carbohydrates and the resulting ruminal VFA concentrations [16]. To understand the effects of grain-rich diets on gene expression in the rumen epithelium, rumen papillae biopsies are commonly used to investigate the aforementioned epithelial functions [17]. Rumen papillae are formed by the ruminal mucosa as surface enlarging structures and are thus composed of a multilayered squamous epithelium with submucosal connective tissue with blood vessels.

However, different sections of the rumen papillae exist in different micromilieus, which may have distinct pH, redox potential, and nutrient availability. For example, the apical section of the rumen papillae is closer to the liquid phase of the rumen, which has a higher pH and lower redox potential than the medial and basal sections, where solid particles and microbial biofilms are more abundant [18]. These differences in physiochemical conditions may also influence gene expression along the papilla axis.

Data concerning differences in the gene expression patterns in the papillae sections remain incipient and are very limited. Since investigations regarding site-specific gene expression analyses on rumen epithelial tissues are lacking, the characterization of the most suitable papilla section for gene expression analyses is still to be defined. This study aimed to compare the expression profiles across various parts of the papillae by analyzing gene expression profiles in different sections of bovine papillae biopsies from cows under SARA conditions. The cows were initially fed a forage-based diet (FD) for one week before transitioning to a high-grain diet (HG, 65% concentrate) to induce SARA. To investigate the gene expression patterns, well-characterized genes involved in barrier function, microbial sensing, innate and adaptive immune responses, and nutrient transport across the rumen epithelium were selected. Quantitative RT-qPCR was performed using RNA extracted from the apical part (AP), basal part (BP), and total length (TP) of the papillae.

## 2. Results

### 2.1. Gene Expression of Epithelial Integrity and Nutrient Transport Genes

The effects of the transition from a FD to HG diet on the relative expression of the barrier function-related genes *DSG1* and *ZO1* were evident in AP only (Figure 1a,b). A significantly higher expression of *DSG1* was observed in the AP samples in wk3 (*p* = 0.0317) when compared to the FD diet, whereas the BP samples only tended to express more *DSG1* between the baseline and wk3 (*p* = 0.0556). A continuous increase in *ZO1* expression over 3 weeks persisted solely in the AP samples, with a significant increase in *ZO1* expression between the baseline and wk1 (*p* = 0.0159) and wk3 (*p* = 0.0159), respectively. In the BP, similar to *DSG1,* only a trend towards an increased expression of *ZO1* after the transition to HG could be determined between the baseline and wk3 (*p* = 0.0556). Hence, the barrier function was found to be significantly affected by the diet. The AP and BP samples, which were homogenized using bead beating and protective lysis buffer, differed in their expression profiles regarding *DSG1* and *ZO1*. The stronger effect in the AP compared to BP could be due to papillae tips being covered by epithelial cells along their entire border, which is partially absent in the sectioned basal parts of papillae samples. Thus, the relative amount of epithelial cells is higher in the AP, while basal parts of papillae and entire-length papilla samples contain more cells from muscles, vessels, and connective tissue. The TP samples did not show any differences in *DSG1* or *ZO1* expression, irrespective of the diet.

The high catabolic activity of rumen microbiota and the typically low concentration of the intermediate substrate D-glucose in the rumen have led to limited attention being paid to ruminal glucose absorption. However, in dairy cows consuming large amounts of grain, 35–65% of the total ingested starch escapes ruminal fermentation, depending on the starch source and processing [19]. Our analysis of nutrient transport across the rumen epithelium focused on the target genes *SGLT1* and *GLUT3*. The results for the AP, BP, and TP samples over the three time points showed that the AP samples tended to express more *SGLT1* after one week of HG (*p* = 0.0635), with a similar trend observed in the BP samples three weeks after transitioning to the HG diet (*p* = 0.0556) (Figure 1c,d). There were no differences for *SGLT1* in the TP samples. Similar to *ZO1*, the relative expression of *GLUT3* was significantly higher in the AP in wk3-biopsies (*p* = 0.0159). Both the BP and TP showed only a trend toward an increased *GLUT3* expression after 3 weeks of HG feeding (*p* = 0.0556, *p* = 0.0556, respectively). The expression profile of the *GLUT3* gene in the TP samples is very interesting in regard to the homogenization method used for those samples. Among our target genes for epithelial integrity and nutrient transport, *GLUT3* is the only gene for which we found at least a trend for expression differences in TP samples. Nevertheless, the BP and TP show similar expression patterns for both genes, regardless of the extraction method.

### 2.2. Gene Expression of Immune Response Genes

Within the candidate genes for the investigation of the innate immune response, the expression values of *TNFA* were not affected by the diet in the three sample types (Figure 2a). *TLR4*, however, was clearly affected by the diet in the AP and BP, while the TP samples had similar expression profiles irrespective of the diet (Figure 2b). *TLR4* gene expression was increased in both the AP and BP samples in wk3 of HG compared to FD (*p* = 0.0635, *p* = 0.0317), and it was also significantly increased in wk3 compared to wk1 in the AP (*p* = 0.0079). Both proinflammatory genes *TNFA* and *IFNG* showed similar expression patterns in the diet-dependent expression analysis in the AP and BP (Figure 2a,c). The TP samples showed an increased *IFNG* expression between wk1 and wk3 of HG feeding (*p* = 0.0317). Especially in these two genes, a quite high variance was observed in the FD samples, suggesting different inflammatory states in the individual animals’ rumen at the end of the baseline feeding period and potentially masking changes in gene expression by HG feeding.

Both immune response-related genes *IFNG* and *NFKB* (Figure 2c,d) were shown to be affected by the diet. While *NFKB* showed a significantly higher expression in wk1 compared to FD in the AP samples (*p* = 0.0159), the TP and BP samples had similar expression profiles irrespective of the diet. Although the expression of *NFKB* appeared to be even higher in wk3 of HG in the AP, the difference was not found to be significant. We found a similar expression profile of *TLR4* and *NFKB* in the AP samples (Figure 2b,d), which is in line with the *TLR4*-stimulated immune response via the NFKB pathway.

### 2.3. Gene Expression at Different Sampling Locations

In order to determine whether the gene expression in the sampling sites differed within the same diet group, the genes that showed the most obvious differences between the papillae sites were tested for differences in sample location. In doing so, we selected the time point where cows were transitioned from FD to the HG diet (=wk1) and tested one gene for each parameter (Figure 3).

*ZO1* showed the lowest expression in the AP, followed by the BP samples. However, a significantly higher expression was determined solely in TP compared to the AP samples (*p* = 0.0079) (Figure 3a). The lowest *SGLT1* expression was found in the BP. Similar to *ZO1*, the highest expression was found in the TP samples (*p* = 0.0317). Moreover, a trend towards an increased expression was seen in the TP compared to the AP (*p* = 0.0952) (Figure 3b). *TLR4* gene expression was significantly higher in the TP compared to the AP (*p* = 0.0079), and a trend towards a higher expression was observed in the TP samples compared to the BP (*p* = 0.0635) (Figure 3c).

## 3. Discussion

The results from our study provide insights into how dietary changes affect gene expression profiles across different regions of the bovine rumen papillae. Our observations reveal distinct regional responses to the high-grain diet and underscore the importance of sampling location in understanding diet-induced physiological changes.

Immunolocalization of *ZO1* in the bovine rumen epithelium *stratum granulosum* and *stratum spinosum* with a decreasing density from apical towards basal papillae, as reported by Graham and Simmons [20], is consistent with our expression data for *ZO1* in the AP and BP samples. These findings highlight that the apical part of the papillae is the most suitable site with the highest sensitivity to analyze diet-induced changes in epithelial barrier integrity. Additionally, *DSG1* showed significant differences between the sampling sites, indicating a higher expression of barrier-forming proteins at the papillae tips in the rumen epithelium. Consequently, the TP samples might not be ideal for this purpose due to their reduced sensitivity.

The results regarding nutrient transport genes suggest that ruminal glucose transport and/or adaptation to HG diets may occur primarily in the apical part of the papillae. Both nutrient transporters were regulated most obviously in the AP samples, with *GLUT3* being upregulated significantly and *SGLT1* by a trend in the AP samples, whereas they showed only a tendency towards increased expression in the BP or TP. Increased *SGLT1* transcription in murine small intestinal crypt epithelial cells is associated with differentiated cell renewal from the basal to apical regions [21], which may also occur in rumen papillae. While ovine small intestinal epithelium does not express *SGLT1* protein or mRNA under normal feeding conditions [22], glucose infusion into the ovine gut lumen leads to a significant increase in *SGLT1* activity and protein abundance [22,23], suggesting that *SGLT1* expression is subject to translational or posttranslational regulation. Supporting this, *SGLT1* mRNA and activity were found in the forestomach of sheep and cows as well as in the bovine intestine [24,25,26], indicating that intraluminal carbohydrates are not fully converted into VFA and that *SGLT1* expression adapts in cattle who are fed grain-rich diets. Although our data showed a tendency towards increased *SGLT1* expression at week 1 after transitioning to a HG diet in the AP samples and after three weeks in the BP samples, the changes were not statistically significant. Considering that the TP showed no expression changes, our findings suggest that, if at all, an adaptive regulation of glucose transporter expression remains limited to the tips of rumen papillae, which underscores the importance of the apical papillae in glucose uptake. However, additional studies are warranted to further investigate this potential adaptive mechanism and confirm the trend observed in SGLT1 expression.

In this study, TLR4 was selected for its role as a key microbial sensor, detecting both lipopolysaccharides from Gram-negative bacteria and endogenous molecules released during tissue injury [27]. Its involvement in the immune response makes TLR4 a promising marker for evaluating proinflammatory responses in cows experiencing SARA. The observed increase in *TLR4* gene expression from wk1 to 3 of HG feeding suggests a link between the progression of SARA and tissue damage, which aligns with previous ruminal pH estimations that confirmed the presence of SARA. Notably, *TLR4* expression increased earlier (from week 1) in the AP biopsies compared to BP, where the increase was only evident by wk3. This suggests that the AP samples may be more sensitive and appropriate for detecting the early inflammatory responses associated with grain-rich diets, potentially offering a more robust model for investigating tissue inflammation. In addition to *TLR4*, the expression of adaptive immune response genes, *IFNG* and *NFKB*, was influenced by diet. Specifically, *NFKB* expression was significantly elevated in the AP samples during wk1 of HG feeding, compared to FD feeding, while the TP and BP samples exhibited consistent expression across both diets. The parallel expression patterns of *TLR4* and *NFKB* in the AP samples support the involvement of the TLR4-mediated immune response via the NFKB pathway. These findings further suggest that the AP samples are preferable for investigating the dietary impacts on *NFKB* expression compared to the BP or whole papillae biopsies.

These results align with our initial assumption that the TP, with its higher connective tissue content, would exhibit fewer changes in epithelial gene expression in response to dietary shifts. Interestingly, we observed a higher gene abundance across all analyzed genes in the TP compared to the AP and BP. This may be explained by the different extraction methods used, as the grinding of the TP likely resulted in more efficient RNA extraction. However, despite the TP yielding a higher total gene copy number, the AP samples consistently proved to be the most sensitive and reliable for detecting dietary-induced changes. The increased responsiveness of the AP samples, particularly in relation to inflammation and immune response, reinforces their value as the optimal tissue for studying the effects of grain-rich diets on gene expression. In future studies, these findings should be considered when choosing the sampling site and extraction method according to the questions to be answered.

In summary, our study highlights the apical part of the rumen papillae as the most responsive and sensitive region to dietary changes, particularly in terms of epithelial integrity, nutrient transport, and immune response. The differential expression of key genes such as *SGLT1*, *GLUT3*, *TLR4*, and *NFKB* across different papillae sites underscores the importance of selecting appropriate sampling locations for investigating dietary effects. The AP samples, with their higher sensitivity and responsiveness, make a good choice for future studies focusing on epithelial function and immune response.

## 4. Materials and Methods

### 4.1. Animals

Papillae biopsies used for this study were obtained during an experiment conducted at the research and training farm of the University of Veterinary Medicine Vienna, Austria. This study was approved by the Austrian national authority according to the law for animal experiments (protocol number: BMNWF-68.205/0003-V/3b/2019). Detailed information on animals, housing conditions, as well as diet composition and the feeding protocol have been published previously [28]. Briefly, five ruminal-cannulated non-lactating Holstein cows (average age of 11.3 ± 2.3 years, body weight of 909.8 ± 98.2 kg) were adapted to the feeders with a forage-only diet for two weeks. Subsequently, they were fed a forage diet (FD) for one week (45% grass silage, 45% corn silage, and 10% grass hay on a dry matter basis). After one week of adaptation, the cows were transitioned to a high-grain diet (HG) (26.25% grass silage, 8.75% corn silage, and 65% concentrate on a dry matter basis) to induce SARA. The ruminal pH was measured every 15 min via a Lethbridge Research Centre Ruminal pH Measurement System (LRCpH; Dascor Inc., Escondido, CA, USA) [28]. SARA conditions were confirmed when the ruminal pH was below 5.8 for at least 320 min/d [18].

### 4.2. Tissue Collection

Rumen papillae samples were collected from the ruminal ventral sac at the end of the FD period, as well as on day seven (wk1) and day 21 (wk3) after the beginning of the HG feeding challenge. The biopsies were taken 4 h after the morning feeding. Therefore, the cannula was opened, the rumen was partially emptied, and a rumen area approximately 20 cm below the fistula was exteriorized through the cannula. In order to remove visible feed particles, the rumen wall was rinsed with phosphate-buffered saline (PBS; pH 7.4). Papillae samples were seized at their upper part with tweezers, cut at their base using scissors (Figure 4a), and immediately submerged in 50 mL of pre-cooled PBS. After washing gently in PBS to rinse the blood and carefully drying the papillae on sterile gauze, the samples were snap-frozen in liquid nitrogen and stored at −80 °C until further processing. Between each sampling procedure, each piece of equipment was washed and disinfected with ethanol 70%.

### 4.3. Histology of Papillae Biopsies

For histology evaluations, ruminal papillae were sampled as described above, washed in PBS, and fixed in 4% formaldehyde for 6 h before transferring the samples to 70% ethanol for storage at 4 °C. Then, the papillae were dehydrated, embedded in paraffin, and cut into 5 µm thin slices. For staining, the slides were deparaffinized and stained with haematoxylin and eosin. Images were taken with a Zeiss Axioskop and an Axiocam 503 (Carl Zeiss, Jena, Germany) using the ZEN microscopy software (ZEN 2.3 lite).

### 4.4. Tissue Homogenization, RNA Isolation, and RNA Integrity and Quantity

This experiment was conducted using only samples with intact papilla lengths. For each cow, we performed RNA extraction and quantitative RT-qPCR from the apical part (AP), basal part (BP), and total papillae (TP), respectively (Figure 4b). For the isolation of RNA from the TP, frozen whole-length papillae were transferred into a pre-cooled mortar containing liquid nitrogen. Using a pestle, the papillae were ground to powder and collected immediately into cryotubes for further storage in liquid nitrogen. A total of 20–25 mg of pulverized tissue was used for further processing. To isolate the RNA from the AP and BP of the papillae, frozen papillae samples were transferred carefully to a sterile Petri dish on ice. For the AP, only the apical third of the total papilla length was cut off from several papillae until a total of 20–25 mg of tissue was reached. The same procedure was performed for the BP by collecting only the basal third of the total papilla length. All samples (AP, BP, TP) were collected into 2 mL sterile bead beating tubes (Sarstedt, Nümbrecht, Germany) filled with approximately 0.6 g of sterile ceramic beads with a diameter of 1.4 mm (VWR, Vienna, Austria) and 350 µL lysis buffer RLT of the RNeasy Mini Qiacube Kit (Qiagen, Hilden, Germany).

All samples were homogenized mechanically using the FastPrep-2 instrument (MP Biomedicals, Santa Ana, CA, USA) for 30 s at 6 m/s, following an incubation on ice for 2 min to allow the samples to cool down. Thereafter, the samples were centrifuged for 1 min at 10,000× *g,* and the lysate was collected in fresh 1.5 mL reaction tubes. After another centrifugation step for 3 min at 14,680× *g*, the supernatant was carefully collected in a fresh 2 mL reaction tube, avoiding any transfer of disrupted tissue pellet. Using the RNeasy Mini Qiacube Kit and the robotic QIACUBE workstation, total RNA was extracted following the manufacturer’s protocol and eluted in 50 µL RNase-free water. Following genomic DNA removal using DNase I (Ambion^®^ TURBO DNA free, Thermo Fisher Scientific, Waltham, MA, USA), the RNA quality was assessed via determination of the RNA integrity using the Qubit RNA IQ Assay Kit and the Qubit RNA HS Assay Kit (Thermo Fisher Scientific, Waltham, MA, USA) for RNA quantity on the Qubit Fluorometer 4.0 (Life Technologies Corporation, CA, USA). RNA integrity numbers for the AP were 8.3–9.8, for BP 8.3–9.7, and for TP 8.5–9.7.

### 4.5. Primers: Selection, Design, and Validation

Primers for the amplification of reference and target genes (Table 1) were either newly designed or recently established for previous projects. Primer design was performed using ensembl (ensemble.org, accessed on 14 February 2022), Reverse Complement (bioinformatics.org, accessed on 14 February 2022), and Primer3 Input (version 0.4.0, accessed on 14 February 2022), targeting a region spanning between two or more exons. All primer sequences were double-checked regarding their specificity using Primer-BLAST (NIH, https://www.ncbi.nlm.nih.gov/tools/primer-blast/, accessed on 21 February 2022). The determination of the optimal annealing temperature for each primer pair and the specificity of the amplified product was performed by running a gradient RT-qPCR, including the creation of melting curves with the settings described below. The annealing temperatures ranged between 57–65 °C. For the investigation of the epithelial barrier function, tight junction genes encoding for desmoglein (*DSG1*) and *zonula occludens* 1 (*ZO1*) were chosen, whereas we selected tumor necrosis factor alpha (*TNFA*) and toll-like receptor 4 (*TLR4*) as candidates for the innate immune response. Analysis of the adaptive immune response was performed using primer pairs for genes encoding for the cytokine interferon gamma (*IFNG*) and the essential inflammatory transcription factor nuclear factor ‘kappa-light-chain-enhancer’ of activated B-cells (*NFKB*). Glucose transport capacities were analyzed using primer pairs for sodium glucose cotransporter 1 (*SGLT1*) and glucose transporter 3 (*GLUT3*). For the normalization of the mRNA content in the samples, hypoxanthine phosphoribosyltransferase 1 (*HPRT1*) and ornithin-decarboxylase-antizym 1 (*OAZ1*) were chosen as reference genes. The stability of the reference genes was checked using RefFinder [29,30].

### 4.6. RT-PCR and RT-qPCR

For the reverse transcription of complementary DNA (cDNA), 10 µL of template containing 1 µg RNA was mixed with 10 µL of the 2 × mastermix in a total volume of 20 µL reaction volume using the High Capacity cDNA Reverse Transcriptase Kit (Applied Biosystems™, Thermo Fisher Scientific, Waltham, MA, USA). The 2 × mastermix consisted of 2 µL of 10 × Buffer, 0.8 µL of 25 × dNTP Mix (100 mM), 2 µL of 10 × RT Random Primers, 1 µL of Multiscribe RT, and 3.7 µL of RNase-free water. A total of 0.5 µL of RNase inhibitor (40,000 U/mL, 20,000 Units, Biozym, Austria) was added to each reaction. Reverse transcription was performed on a thermocycler (Nexus, Eppendorf SE, Hamburg, Germany) under the following conditions: primer annealing at 25 °C for 10 min, DNA polymerization at 37 °C for 2 h, enzyme deactivation at 85 °C for 5 min, and cooling at 4 °C. In order to reach a final concentration of 10 ng/µL in each cDNA sample, all samples were diluted with 80 µL of RNase-free water after cDNA synthesis.

Gene expression analysis was performed on a qTOWER 384 (Analytik Jena, Jena, Germany) thermocycler using 384 well plates (Sarstedt, Nümbrecht, Germany) after validation of the primer pairs regarding their product specificity. Each reaction contained 20 ng of cDNA as the template, 80 nM forward primer, 80 nM reverse primer, 5 µL of the 2 × mastermix (Blue S’Green qPCR Kit Separate ROX, Biozym Biotech Trading GmbH, Vienna, Austria), and 2.2 µL of RNase-free water in a total volume of 10 µL. The amplification protocol comprised 3 min at 95 °C for initial denaturation, followed by 40 cycles of 95 °C for 5 s and annealing for 30 s at the estimated optimal annealing temperature for each primer pair. Melting curve analyses were performed with increments of 0.5 °C/s between 55 and 95 °C to analyze the amplification product’s specificity and dimer formation. Negative controls and reverse transcription controls (RT minus) were included in each assay to exclude contaminations, and all samples were run in two technical replicates.

For the calculation of the relative expression of the target genes, the geometric mean of both reference genes *HPRT1* and *OAZ1* was used for the normalization of the RNA content in each sample. Changes in the target gene expression in the AP, BP, and TP over the FD period, wk1 and wk3 HG were analyzed as the fold change using the 2^−∆∆Ct^ method [36]. Briefly, the ΔCt values of target genes were determined by the subtraction of the geometric mean of the cycle threshold (Ct) of the reference genes from the Ct of the target genes. For each papillae site (AP, BP, TP), the mean of the ΔCt values for the FD diet, as the baseline, from the TP were calculated and then subtracted from the ΔCt values of the AP, BP, and TP for wk1 and wk3 of the HG diet to assess the ΔΔCt per papillae site. The relative gene expression per target gene is shown as 2^−∆∆Ct^.

### 4.7. Statistical Analysis

Statistical differences in target gene expression among AP, BP, and TP within the FD period, as well as in weeks 1 and 3 of the HG period, were analyzed using the Mann-Whitney U test. Furthermore, differences in gene expression between the FD period (baseline), wk1, and wk3 were evaluated for the AP, BP, and TP using a Mann–Whitney U test. All tests were performed using GraphPad Prism version 8.0.2 for Windows. Differences between groups were considered statistically significant at *p* < 0.05 (* = *p* < 0.05, ** = *p* < 0.01). A statistical tendency is shown as + if *p* < 0.1.

## 5. Conclusions

Our study clearly shows the apical parts of rumen papillae biopsies (AP) to be the most suitable papilla section for gene expression analyses to investigate the diet-induced effects on epithelial integrity, nutrient transport, and immune response. On the contrary, when different sampling locations were compared within the same diet group, the TP samples showed the highest transcript levels of all genes analyzed in this study. Therefore, it is important to carefully consider not only the sampling strategy but also the RNA extraction method used when conducting research on the rumen microbiome and its interactions with the host. Accounting for spatial variations in rumen physiology and microbiology enables researchers to generate more accurate and representative data, reflecting the intricate nature of the ruminal ecosystem.

## Figures and Tables

**Figure 1 ijms-25-12303-f001:**
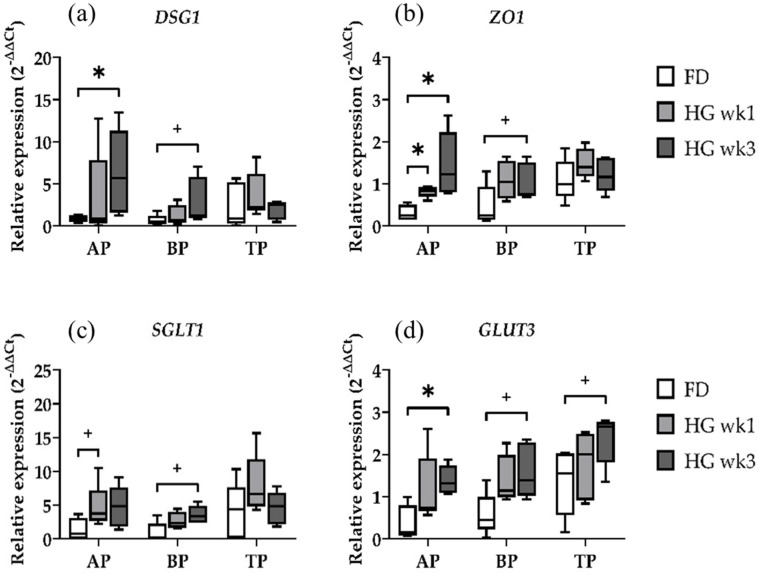
Diet effects on relative gene expression for the epithelial barrier-related genes DSG1 (**a**) and ZO1 (**b**) and the transport-related genes SGLT1 (**c**) and GLUT3 (**d**) in the apical part (AP), basal part (BP), and total length (TP) of the papilla during forage-based diet (FD), and one week and three weeks of high-grain diet (HG). Significance is shown as *, *p* < 0.05. A statistical trend is shown as + when 0.05 < *p* < 0.1.

**Figure 2 ijms-25-12303-f002:**
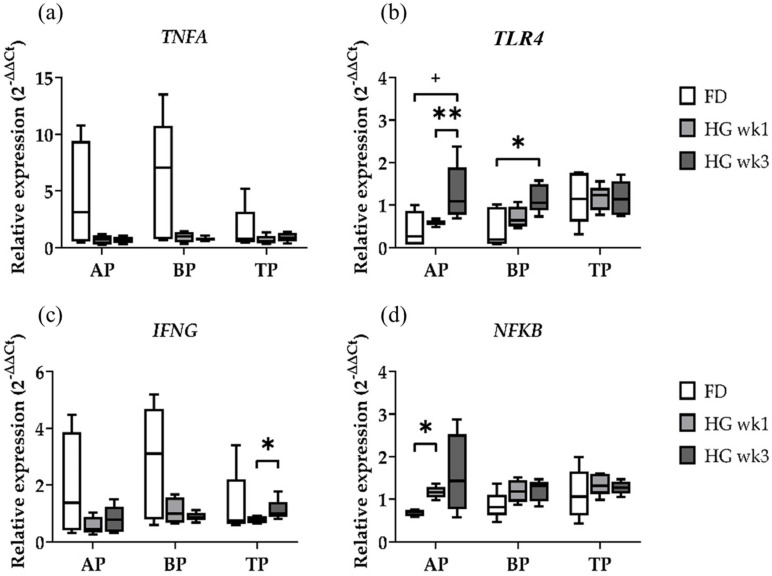
Diet effects on relative gene expression of immune response-related genes *TNFA* (**a**)*, TLR4* (**b**), *IFNG* (**c**), and *NFKB* (**d**) in the apical part (AP), basal part (BP), and total length (TP) of the papilla from forage-based diet (FD) and one week and three weeks after high-grain diet (HG). Significance is shown as *, *p* < 0.05; **, *p* < 0.01. A statistical trend is shown as + if *p* < 0.1.

**Figure 3 ijms-25-12303-f003:**
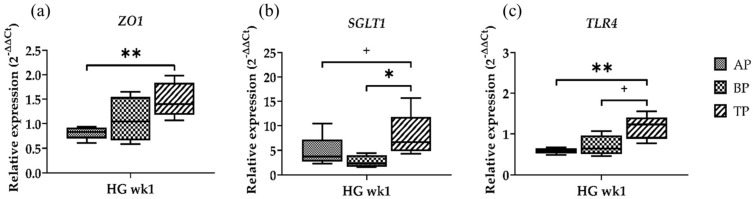
Relative gene expression of *ZO1* (**a**), *SGLT1* (**b**), and *TLR4* (**c**) in the apical part (AP), basal part (BP), and total length (TP) of the papilla one week after high-grain diet. Significance is shown as *, *p* < 0.05; **, *p* < 0.01. A statistical trend is shown as + if *p* < 0.1.

**Figure 4 ijms-25-12303-f004:**
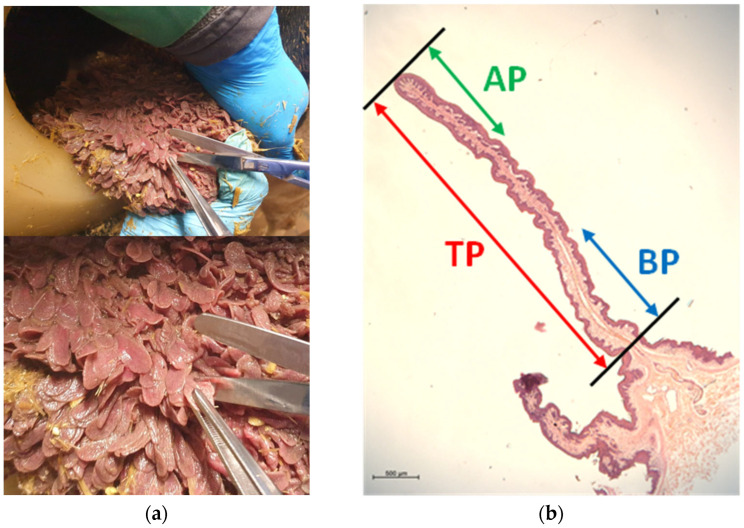
(**a**) Sampling of papillae from a ruminal-cannulated cow. (**b**) H&E staining of a full ruminal papilla, indicating the regions chosen for analysis: the green arrow indicates AP (apical part of papilla), the blue arrow indicates BP (basal part of papilla), and the red arrow indicates TP (total papilla length). Scale bar: 500 µm.

**Table 1 ijms-25-12303-t001:** Oligonucleotide primer sequences used in quantitative PCR in different sites of bovine papillae biopsies.

Gene	Accession Number	Primer Sequence (5′ to 3′)		Annealing Temperature (°C)	Amplicon Size (bp)	References
*Reference genes*						
*HPRT1*	NW_005397637.1	F:	TTGTATACCCAATCATTATGCTGAG	58	109	This study
		R:	ACCCATCTCCTTCATCACATCT			
*OAZ1*	NM_001127243.2	F:	CACAAGAACCGTGATGATCGA	58	69	Petri et al., 2018 [31]
		R:	TCTCACAATCTCAAAGCCCAAA			
*Epithelial barrier function*						
*DSG1*	NM_174045.1	F:	AGACAGAGAGCAATATGGCCAGT	60	88	Steele et al., 2010 [32]
		R:	TTCACACTCTGCTGACATACCATCT			
*ZO1*	XM_024982012.1	F:	CGACCAGATCCTCAGGGTAA	60	161	Liu et al., 2013 [33]
		R:	AATCACCCACATCGGATTCT			
*Innate immune response*						
*TNFA*	NM_173966.3	F:	AGCCCTCTRGTTCARACACT	58	159	This study
		R:	GCTGGTTGTCTTCCAGCTTC			
*TLR4*	XM_005210586.3	F:	TGGGACCCTTGCGTACAG	60	159	This study
		R:	ACGGCCACCAGCTTCTG			
*Adaptive immune response*						
*IFNG*	NM_174086.1	F:	GCAGCTCTGAGAAACTGGAGGA	58	79	This study
		R:	ATGGCTTTGCGCTGGATCT			
*NFKB*	NM_001076409.1	F:	ATACGTCGGCCGTGTCTAT	58	144	Jin et al., 2016 [34]
		R:	GGAACTGTGATCCGTGTAG			
*Glucose transport*						
*SGLT1*	XM_005218131.3	F:	ACATCGCCTACCCGACCTT	64	74	Metzler-Zebeli et al., 2014 [35]
		R:	CAGCATGACCGACAGCATCA			
*GLUT3*	NM_174603.3	F:	ATGTCGCAGGAGAAGCAAGT	64	64	This study
		R:	AGACACCACGGTGAAGATG			

## Data Availability

Data is contained within the article.

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
