# Peer review of "Site Matters: Differences in Gene Expression Profiles Along the Bovine Rumen Papilla During Subacute Rumen Acidosis"

_ijms, 2024, doi:10.3390/ijms252212303_

Round 1
Reviewer 1 Report
Comments and Suggestions for Authors
The article effectively addresses a critical issue in ruminant nutrition: subacute rumen acidosis (SARA). The abstract succinctly summarizes the study's objective, which aims to understand how the sampling site along the ruminal papilla influences gene expression profiles during SARA, thereby highlighting the relevance of the topic. The introduction provides a comprehensive overview of the complex nature of the rumen environment, emphasizing the critical role of microbial consortia in fermentation processes. It effectively establishes the context for understanding ruminal functionality, energy supply, and the overall health of ruminants. The study's objective is clearly defined, focusing on the analysis of gene expression profiles in different sections of the rumen papillae to assess their contributions to epithelial function, particularly during SARA. However, a clearly articulated hypothesis should be included. The methodology section is well-structured and describes in detail the techniques used in the study. The results are well-presented and discussed, but the discussion presents several findings that require a more cautious interpretation. For instance, statements indicating that "changes were not statistically significant" (lines 176-177) and that "NFKB expression also increased... but this change did not reach statistical significance" (lines 196-197) should be framed with greater care. It is essential to discuss the potential implications of the results more thoroughly.
Author Response
Dear reviewer,
thank you very much for your effort reviewing our manuscript and for your appreciation of our work.
According to your valuable suggestions, we tried to emphasize the aim of the study better (l. 59). Since this study was mainly intended to improve methodology and reveal differences between different sampling locations along a ruminal papilla, we feel that formulating an aim is more appropriate than a hypothesis and hope that you agree with that.
Furthermore, we thank you for your comment regarding the discussion. We changed it towards a more careful wording and avoiding an overinterpretation of statistical trends and deleted therefore as well the sentence “NFKB expression was also increased in AP samples by wk 3, but this change did not reach statistical significance.”
We elaborated further on potential implications of our findings for future studies in l. 212ff.
We hope that we understood your remarks correctly and you find our changes improved the manuscript. Thank you again for reviewing our manuscript.
Reviewer 2 Report
Comments and Suggestions for Authors
The paper investigates the impact of diet-induced changes on gene expression in different sections of bovine rumen papillae, focusing on epithelial integrity, nutrient transport, and immune response. It emphasizes the importance of sampling strategy and spatial variation for accurate representation of ruminal physiology and microbiology.
Overall, the paper demonstrates a solid experimental design with clear objectives and detailed methodology. However, it needs some refinements in terminology and interpretation.
Key issues and inconsistencies include:
Lines 2-3: I recommend the authors consider revising the title for a more formal and precise description of the study, such as: “Variations in Gene Expression Profiles Along the Different Parts of the Bovine Rumen Papilla in Subacute Ruminal Acidosis.”
Line 13: Change “rumen papilla biopsies” to “ruminal papilla biopsies.”
Line 73: The p-value for DSG1 in BP is reported as p = 0.0556, which is not typically considered statistically significant (p < 0.05). It should be clarified that this reflects a trend rather than a significant result, similar to how other trends are described in this section (e.g., p = 0.0635 on Line 93).
Lines 179-180: The phrase “our findings suggest an adaptive regulation of SGLT1 expression in the tips of rumen papillae” overgeneralizes the data, which only shows a "tendency towards increased SGLT1 expression" without statistically significant results. This should be framed as a potential trend requiring further investigation. I suggest revising the text to acknowledge the lack of statistical significance and highlight the need for additional studies to confirm this potential adaptive regulation.
Lines 203-207: The statement, "contrary to our initial assumption that TP, with its higher connective tissue content, would exhibit fewer changes in epithelial gene expression... we observed higher gene abundance across all analyzed genes in TP," contradicts the earlier conclusion that TP samples are less sensitive (e.g., line 162). The explanation regarding RNA extraction efficiency does not adequately clarify why gene abundance is higher in TP while it is deemed less responsive. Please clarify this inconsistency by explaining whether the higher gene abundance is biologically meaningful or a result of methodological differences. Additionally, explicitly state whether this finding challenges previous assumptions about TP sensitivity.
Lines 209-211: The sentence, "the increased responsiveness of AP samples, particularly in relation to inflammation and immune response, reinforces their value," overlooks the earlier observation that TP samples had a higher gene abundance across all analyzed genes (line 204). This suggests a contradictory relationship between gene abundance in TP samples and their ability to reflect inflammatory responses. Acknowledge the higher gene abundance observed in TP samples and clarify whether this impacts the conclusion that AP samples are the most reliable for detecting inflammation.
Line 357: The statement "TP samples showed the highest expression levels in general" lacks specificity. It should clarify which gene or genes had the highest expression levels. Since the study involves multiple genes related to various pathways (nutrient transport, immune response, epithelial integrity), it would be clearer to mention whether this refers to a specific gene subset or overall expression trends across different genes. More precision is needed to avoid confusion.
Line 359: The term "extraction method" is somewhat vague. Since the methods section discusses both RNA extraction and tissue sampling techniques, it would be helpful to specify which extraction method is being referred to—whether it pertains to RNA extraction or tissue sampling. The lack of clarity may confuse readers regarding the aspect of the methodology being emphasized here.
Author Response
Dear Reviewer,
thank you very much for your carful review of our manuscript. We appreciate your suggestions. Please find a point-by-point reply to your comments below.
Key issues and inconsistencies include:
Lines 2-3: I recommend the authors consider revising the title for a more formal and precise description of the study, such as: “Variations in Gene Expression Profiles Along the Different Parts of the Bovine Rumen Papilla in Subacute Ruminal Acidosis.”
Response: Thank you very much for your thoughtful suggestion regarding the title. We appreciate your perspective and understand the value of a more formal title structure. However, after careful consideration, we feel that our chosen title provides a balance between scientific rigor and reader engagement. While we understand that using "different parts of the bovine rumen papilla" may add specificity, we are concerned that it might result in a longer title that could potentially reduce readability or introduce ambiguity. We hope that our approach will still meet the expectations of precision and clarity for the study. Thank you again for your valuable input.
Line 13: Change “rumen papilla biopsies” to “ruminal papilla biopsies.”
Response: We corrected that.
Line 73: The p-value for DSG1 in BP is reported as p = 0.0556, which is not typically considered statistically significant (p < 0.05). It should be clarified that this reflects a trend rather than a significant result, similar to how other trends are described in this section (e.g., p = 0.0635 on Line 93).
Response: Thank you for your careful reading and valuable feedback on the interpretation of the p-values. We appreciate your point regarding the threshold for statistical significance. In the manuscript, we have been mindful to describe p-values above 0.05 as trends rather than significant results, using wording such as “tended” to clarify these cases. This includes instances such as p = 0.0556 and p = 0.0635
Lines 179-180: The phrase “our findings suggest an adaptive regulation of SGLT1 expression in the tips of rumen papillae” overgeneralizes the data, which only shows a "tendency towards increased SGLT1 expression" without statistically significant results. This should be framed as a potential trend requiring further investigation. I suggest revising the text to acknowledge the lack of statistical significance and highlight the need for additional studies to confirm this potential adaptive regulation.
Response: Thank you for highlighting this point. We appreciate your suggestion to avoid overgeneralizing our findings on SGLT1 expression. To address this, we have revised the text to clarify that our findings indicate a limited adaptive regulation in the tips of rumen papillae and now include an additional sentence to emphasize the need for further research to confirm this potential trend. Thank you again for your helpful input.
We wrote now: “Considering that TP showed no expression changes, our findings suggest that, if at all, an adaptive regulation of glucose transporter expression remains limited to the tips of rumen papillae, underscoring the importance of the apical papillae in glucose uptake. However, additional studies are warranted to further investigate this potential adaptive mechanism and confirm the trend observed in SGLT1 expression.”
Lines 203-207: The statement, "contrary to our initial assumption that TP, with its higher connective tissue content, would exhibit fewer changes in epithelial gene expression... we observed higher gene abundance across all analyzed genes in TP," contradicts the earlier conclusion that TP samples are less sensitive (e.g., line 162). The explanation regarding RNA extraction efficiency does not adequately clarify why gene abundance is higher in TP while it is deemed less responsive. Please clarify this inconsistency by explaining whether the higher gene abundance is biologically meaningful or a result of methodological differences. Additionally, explicitly state whether this finding challenges previous assumptions about TP sensitivity.
Response: Thank you for your thoughtful feedback on this point. We understand your concern regarding the potential inconsistency between gene abundance in TP samples and their observed responsiveness. To clarify, we believe that the higher gene abundance in TP samples likely reflects methodological differences, particularly the more efficient RNA extraction due to grinding. Despite this higher gene copy number, TP samples consistently demonstrated the lowest responsiveness to dietary changes, supporting our initial assumption about their reduced sensitivity in epithelial gene expression. We have revised the text to highlight this point, aiming to clarify that the observed abundance in TP does not contradict our findings regarding TP sensitivity. Thank you for prompting us to address this aspect more explicitly.
We wrote now “These results align with our initial assumption that TP, with its higher connective tissue content, would exhibit fewer changes in epithelial gene expression in response to dietary shifts. Interestingly, we observed higher gene abundance across all analyzed genes in TP compared to AP and BP. This may be explained by the different extraction methods used, as the grinding of TP likely resulted in more efficient RNA extraction. However, de-spite TP yielding a higher total gene copy number, AP samples consistently proved to be the most sensitive and reliable for detecting dietary-induced changes.” (lines 210 and following)
Lines 209-211: The sentence, "the increased responsiveness of AP samples, particularly in relation to inflammation and immune response, reinforces their value," overlooks the earlier observation that TP samples had a higher gene abundance across all analyzed genes (line 204). This suggests a contradictory relationship between gene abundance in TP samples and their ability to reflect inflammatory responses. Acknowledge the higher gene abundance observed in TP samples and clarify whether this impacts the conclusion that AP samples are the most reliable for detecting inflammation.
Response: Thank you for your thoughtful feedback on this point. We understand your concern regarding the potential inconsistency between gene abundance in TP samples and their observed responsiveness. To clarify, we believe that the higher gene abundance in TP samples likely reflects methodological differences, particularly the more efficient RNA extraction due to grinding. Despite this higher gene copy number, TP samples consistently demonstrated the lowest responsiveness to dietary changes, supporting our initial assumption about their reduced sensitivity in epithelial gene expression. We have revised the text to highlight this point, aiming to clarify that the observed abundance in TP does not contradict our findings regarding TP sensitivity. Thank you for prompting us to address this aspect more explicitly.
We wrote now “These results align with our initial assumption that TP, with its higher connective tissue content, would exhibit fewer changes in epithelial gene expression in response to dietary shifts. Interestingly, we observed higher gene abundance across all analyzed genes in TP compared to AP and BP. This may be explained by the different extraction methods used, as the grinding of TP likely resulted in more efficient RNA extraction. However, de-spite TP yielding a higher total gene copy number, AP samples consistently proved to be the most sensitive and reliable for detecting dietary-induced changes.” (lines 210 and following)
Line 357: The statement "TP samples showed the highest expression levels in general" lacks specificity. It should clarify which gene or genes had the highest expression levels. Since the study involves multiple genes related to various pathways (nutrient transport, immune response, epithelial integrity), it would be clearer to mention whether this refers to a specific gene subset or overall expression trends across different genes. More precision is needed to avoid confusion.
Response: Thank you for your suggestion to improve the specificity of this statement. In response, we have rephrased the sentence to read, “TP samples showed the highest transcript levels of all genes analyzed in this study,” to clarify that this observation applies to all genes included in our analysis. We appreciate your input in helping us make this point clearer.
Line 359: The term "extraction method" is somewhat vague. Since the methods section discusses both RNA extraction and tissue sampling techniques, it would be helpful to specify which extraction method is being referred to—whether it pertains to RNA extraction or tissue sampling. The lack of clarity may confuse readers regarding the aspect of the methodology being emphasized here.
Response: To enhance clarity, we have revised the sentence to: “Therefore, it is important to carefully consider not only the sampling strategy but also the RNA extraction method used.” We hope this revision will help readers understand that we are specifically referring to the RNA extraction process. Thank you for helping us improve the precision of our wording.